# COVID-19-Related Factors Associated with Sleep Disturbance and Suicidal Thoughts among the Taiwanese Public: A Facebook Survey

**DOI:** 10.3390/ijerph17124479

**Published:** 2020-06-22

**Authors:** Dian-Jeng Li, Nai-Ying Ko, Yi-Lung Chen, Peng-Wei Wang, Yu-Ping Chang, Cheng-Fang Yen, Wei-Hsin Lu

**Affiliations:** 1Graduate Institute of Medicine, College of Medicine, Kaohsiung Medical University, Kaohsiung 80708, Taiwan; u108800004@kmu.edu.tw (D.-J.L.); wistar.huang@gmail.com (P.-W.W.); 2Department of Addiction Science, Kaohsiung Municipal Kai-Syuan Psychiatric Hospital, Kaohsiung 80276, Taiwan; 3Department of Nursing, College of Medicine, National Cheng Kung University, Tainan 70101, Taiwan; nyko@mail.ncku.edu.tw; 4Department of Healthcare Administration, Asia University, Taichung 41354, Taiwan; elong@asia.edu.tw; 5Department of Psychology, Asia University, Taichung 41354, Taiwan; 6Department of Psychiatry, Kaohsiung Medical University Hospital, Kaohsiung 80708, Taiwan; 7School of Nursing, The State University of New York, University at Buffalo, New York, NY 14214-3079, USA; yc73@buffalo.edu; 8Department of Psychiatry, Ditmanson Medical Foundation Chia-Yi Christian Hospital, Chia-Yi City 60002, Taiwan; 9Department of Senior Citizen Service Management, Chia Nan University of Pharmacy and Science, Tainan 71710, Taiwan

**Keywords:** COVID-19, sleep disturbance, suicidal thoughts, social activities, predictors

## Abstract

Coronavirus disease 2019 (COVID-19) pandemic has impacted many aspects of people’s lives all over the world. This Facebook survey study aimed to investigate the COVID-19-related factors that were associated with sleep disturbance and suicidal thoughts among members of the public during the COVID-19 pandemic in Taiwan. The online survey recruited 1970 participants through a Facebook advertisement. Their self-reported experience of sleep disturbance and suicidal thoughts in the previous week were collected along with a number of COVID-19-related factors, including level of worry, change in social interaction and daily lives, any academic/occupational interference, levels of social and specific support, and self-reported physical health. In total, 55.8% of the participants reported sleep disturbance, and 10.8% reported having suicidal thoughts in the previous week. Multiple COVID-19-related factors were associated with sleep disturbance and suicidal thoughts in the COVID-19 pandemic. Increased worry about COVID-19, more severe impact of COVID-19 on social interaction, lower perceived social support, more severe academic/occupational interference due to COVID-19, lower COVID-19-specified support, and poorer self-reported physical health were significantly associated with sleep disturbance. Less handwashing, lower perceived social support, lower COVID-19-specified support, poorer self-reported physical health, and younger age were significantly associated with suicidal thoughts. Further investigation is needed to understand the changes in mental health among the public since the mitigation of the COVID-19 pandemic.

## 1. Introduction

### 1.1. Ongoing Threat of COVID-19

Coronavirus disease 2019 (COVID-19) is a novel infectious disease that emerged in Wuhan, China at the end of 2019 and rapidly spread worldwide [1]. As the threat increased, the World Health Organization (WHO) declared the outbreak of COVID-19 a global public health emergency on the 30 January 2020 and it was declared a pandemic on the 11 March 2020. The COVID-19 pandemic is severely affecting people’s daily lives all over the world, including their health, economic wellbeing and social interactions [2]. Social distancing, self-isolation, and travel restrictions have resulted in a massive decrease in productivity across all economic sectors, which has had severe lockdown effects and placed a heavy burden on society, such as the decline in global stock markets and overload of healthcare services [3].

People in Taiwan also suffered from threats of COVID-19. The Taiwan Centers for Disease Control (Taiwan CDC) identified the risk of the COVID-19 pandemic and delivered necessary policy quickly at the end of January 2020. According to the situation report of Taiwan CDC [4], 443 cases were infected with COVID-19 at June 12th. Of the confirmed cases, there have been seven deaths, and 431 patients have been released from isolation. Most of the morality was associated with elderly and chronic systemic disease. Healthcare workers followed the order of authorities to increase the intensity of infection control, such as forced wearing of masks, taking body temperatures, and limitations for visitors in hospitals. Due to limited cases of infection, the intense care facilities were sufficient. The relatively minimal proportion of less confident individuals may result from timely border control, application of big data analytics, and experienced teams of officials [5].

### 1.2. Impact of COVID-19 Pandemics on Mental Health

In addition to the socio-economic burden that COVID-19 has caused, the pandemic may also have hazardous effects on the public’s mental health. Several studies had addressed the psychological impact of COVID-19. A web-based survey reported that the overall prevalence of anxiety symptoms, depressive symptoms, and poor sleep quality were 35.1%, 20.1%, and 18.2%, respectively, among the public affected by the COVID-19 outbreak in China [6]. Another study recruited 1257 subjects reported that a considerable proportion of participants reported symptoms of depression (50.4%), anxiety (44.6%), insomnia (34.0%), and distress (71.5%) [7]. It is important to examine COVID-19-related factors that affect mental health during the COVID-19 pandemic. Studies predicting factors associated with mental health problems will enhance understanding and help develop timely prevention and intervention strategies to enhance public mental health during COVID-19 pandemics [8]. It was reported that nurses, women, frontline health care workers, and those working in Wuhan, China were significantly associated with more severe degrees of depression and anxiety symptoms than other health care workers [7]. Another COVID-19 study indicated that female gender, student status, specific physical symptoms (e.g., myalgia or dizziness), and poor self-rated health status were significantly associated with higher levels of stress, anxiety, and depression [9]. In addition, individuals with frequent social media exposure were positively associated with higher odds of anxiety than lower exposed ones during the COVID-19 pandemic [10].

### 1.3. Aim of Current Study

Although previous studies explored the high prevalence of psychological distress for the public during the COVID-19 pandemic, it was still insufficient for investigations regarding the risk factors associated with mental health problems, such as social support. An Egyptian population-based study demonstrated that only 24.2% of individuals had increased support from friends, while 40.6% of them reported increased support from family members during the COVID-19 pandemic [11]. However, whether social support can buffer the negative effects of social isolation and changes in daily lives during the COVID-19 pandemic warrants further study. In addition, seldom COVID-19 related studies discussed suicide, which is an unneglectable issue due to elevated psychological distress [12].

Therefore, the aim of the current study focused on the impact of COVID-19 on daily lives and how varying levels of support may affects people’s mental wellbeing, such as level of sleep disturbance and suicidal thoughts.

## 2. Methods

### 2.1. Participants

Participants were recruited through a Facebook advertisement from April 10 to 23, 2020. Facebook users were eligible for this study if they were ≥20 years old and living in Taiwan. The Facebook advertisement included a headline, main text, pop-up banner, and link to the research website and questionnaire. The advertisement was designed to appear in the Facebook users’ “news feeds,” which is a continually updated list of posts from advertisers and the user’s connections (such as friends and the Facebook groups that they have joined). Our advertisement only targeted users’ news feeds, as opposed to other Facebook advertising locations (e.g., the right column), because news feed advertisements are most effective in recruiting research participants [13]. The advertisement was targeted to Facebook users by location (Taiwan) and language (Chinese), and Facebook’s advertising algorithm determined which users to show our advertisement to. To ensure that health care workers were recruited, the Facebook advertisement was also posted link to LINE (a direct messaging app) and Facebook groups joined by healthcare workers.

This study was approved by the Institutional Review Board (IRB) of Kaohsiung Medical University Hospital (approval no. KMUHIRB-EXEMPT (I) 20200011). As participation was voluntary and survey responses were anonymous, the IRB ruled that this study did not require informed consent. The study participants were given no incentive or reward for their participation. The links were provided regarding the COVID-19 information by the Taiwan CDC, Kaohsiung Medical University Hospital, and Medical College of National Cheng Kung University so participants could learn more about COVID-19.

### 2.2. Questionnaire

#### 2.2.1. Sleep Disturbance and Suicidal Thoughts

The sleep disturbance and suicidal thoughts of participants were assessed with the question on a 5-point Likert scale, with scores ranging from 0 (never) to 4 (extremely severe). Participants who rated the items of sleep disturbance and suicidal thoughts > 0 were classified as having sleep disturbance or suicidal thoughts, respectively [14]. The details of this and following questionnaires are provided as Appendix A.

#### 2.2.2. Demographic Variables

The participants’ gender (female, male, or transgender), age, and education level (high school or below vs. college or above) were collected. Moreover, whether the participants were healthcare workers was also identified.

#### 2.2.3. Worry about COVID-19

The five-item questionnaire developed by Liao and colleagues to assess individuals’ level of worry about H1N1 [15] was applied to assess the severity of participants current worry about COVID-19. Participants responded to them with four 5-point and one 10-point Likert scale. Cronbach’s α was 0.71. A higher total score of the five questions indicated more severe worry about COVID-19 (Appendix A).

#### 2.2.4. Impact of the COVID-19 Pandemic on Participants Daily Lives

With reference to a previous study, seven questions were developed to evaluate the changes to participants daily lives due to COVID-19 [15]. The responses were transformed into 0 (“no” or “yes, but not due to COVID-19”) and 1 (“yes, due to COVID-19”). Cronbach’s α of the questionnaire was 0.57 (Appendix A).

#### 2.2.5. Impact of the COVID-19 Pandemic on Social Interaction

Four questions were developed to assess the impact of COVID-19 on social interaction. The responses were transformed into 0 (“no” or “yes, but not due to COVID-19”) and 1 (“yes, due to COVID-19”). Cronbach’s α was 0.78. Higher total scores for the four questions represented COVID-19 having a more severe impact on the individual’s social interactions (Appendix A).

#### 2.2.6. Academic/Occupational Interference by the COVID-19 Pandemic

The academic/occupational interference was evaluated by the COVID-19 pandemic with one question. Participants responded to the question on a 5-point Likert scale ranging from 0 (entirely no interference) to 4 (extreme interference) (Appendix A).

#### 2.2.7. Perceived Social Support and COVID-19-Specified Support

Three questions were used to measure the level of perceived social support. The responses to these three items were graded on a 5-point Likert scale, with scores ranging from 0 (entirely disappointed) to 4 (extremely satisfied). Cronbach’s α was 0.81. A higher total score represented a higher level of perceived social support.

The participants’ level of COVID-19-specified support was estimated with five questions. The responses to the questions were graded on a 3-point Likert scale, ranging from 0 (entirely insufficient), 1 (mild insufficient), and 2 (sufficient). Cronbach’s α was 0.75. A higher total score represented higher satisfaction with the level of COVID-19-specified support (Appendix A).

#### 2.2.8. Self-Reported Physical Health

Two questions from the Self-Perceived Health Questionnaire developed by Ko and colleagues [16] were used to assess the level of self-reported physical health. The level of self-reported physical health was a 5-Likert scale rated from 1 (much worse) to 5 (much better). Cronbach’s α was 0.81. A higher total score indicated better self-reported physical health (Appendix A).

### 2.3. Statistical Analysis

Descriptive analysis was used to summarize the variables. Univariate logistic regression with crude odds ratios (cOR) was used to identify potential COVID-19-related factors associated with sleep disturbance and suicidal thoughts. Furthermore, all potential predictive variables identified from the first step were eligible for inclusion in the forward stepwise logistic regression models with adjusted odds ratios (aOR) to determine the independent predictors for sleep disturbance and suicidal thoughts. All tests were examined using a two-tailed test with the alpha level set at <0.05. All data were processed using SPSS version 23.0 for Windows (SPSS Inc., Chicago, IL, USA).

## 3. Results

### 3.1. Summary of Descriptive Analysis

Initially, there were 2031 respondents who filled in the online questionnaire. After excluding those respondents with missing values (*n* = 31) and those aged below 20 years (*n* = 30), the data of 1970 participants (1305 females, 650 males, and 15 transgender) were entered for analysis. The mean age of the participants was 37.81 ± 11.00 years. A summary of the remaining characteristics for all participants are listed in Table 1 and Table 2.

### 3.2. Predictors of Sleep Disturbance and Suicidal Thoughts

A total of 1099 (55.8%) participants reported sleep disturbance, and 212 (10.8%) reported having suicidal thoughts in the past week (Table 3). Table 1 and Table 2 show the results of the univariate logistic regression. The results demonstrated that current sleep disturbance was significantly associated with several factors, including younger age (cOR = 0.99; *p* = 0.002), more severe worry about COVID-19 (cOR = 1.07; *p* < 0.001), more severe impact of COVID-19 on social interaction (cOR = 1.14; *p* = 0.001), lower perceived social support (cOR = 0.87; *p* < 0.001), lower self-reported physical health (cOR = 0.76; *p* < 0.001), higher academic/occupational interference (cOR = 1.20; *p* < 0.001), lower COVID-19-specified support (cOR = 0.85; *p* < 0.001), and more household disinfection (cOR = 1.23; *p* = 0.03).

Table 4 shows the results of the forward stepwise logistic regression. The results indicated that more severe worry about COVID-19 (aOR = 1.04; *p* = 0.001), more severe impact of COVID-19 on social interaction (aOR =1.07; *p* = 0.047), lower perceived social support (OR = 0.91; *p* < 0.001), poorer self-reported physical health (aOR = 0.80; *p* < 0.001), more severe impact of COVID-19 on academic/occupational interference (aOR = 1.12; *p* = 0.005), and lower COVID-19-specified support (OR = 0.92; *p* = 0.003) remained significant independent predictors associated with sleep disturbance.

For suicidal thoughts, the univariate logistic regression indicated several potential factors that could predict suicidal thoughts, including a younger age (cOR = 0.93; *p* < 0.001), being transgender (cOR = 5.55; *p* = 0.001), being non-healthcare workers (cOR = 0.41; *p* < 0.001), more severe worry about COVID-19 (cOR = 1.03; *p* = 0.032), lower perceived social support (cOR = 0.73; *p* < 0.001), poorer self-reported physical health (cOR = 0.72; *p* < 0.001), lower COVID-19-specified support (cOR = 0.76; *p* < 0.001), and less handwashing (cOR = 0.48; *p* = 0.001) (Table 2 and Table 3).

The results of the forward stepwise logistic regression demonstrated that younger age (aOR = 0.93; *p* < 0.001), less handwashing (aOR = 0.48; *p* = 0.002), lower perceived social support (aOR = 0.79; *p* < 0.001), poorer self-reported physical health (aOR = 0.81; *p* < 0.001), and lower COVID-19-specified support (OR = 0.81; *p* < 0.001) remained significant independent predictors of suicidal thoughts (Table 4).

## 4. Discussion

The findings of the current study were discussed initially with the primary outcome, including sleep and suicidal thoughts. Then several independent predictors associated with sleep disturbance and suicidal thoughts were discussed. The current study was conducted using an online survey, which has been reported to be a promising method for evaluating how the general public understand and perceive a fast-moving infectious disease outbreak [17]. In the current study, the rate of sleep disturbance in the previous week was 55.8%, which was much higher than the result of a population-based investigation in Taiwan using the same question to assess sleep disturbance (up to 28.3%) [14]. The rate of suicidal thoughts reached 10.8% in the previous week, which was also higher than the findings of a previous survey in Taiwan using the same question to assess suicidal thoughts (2.1%) [14]. In addition, increased worry about COVID-19, more severe impact of COVID-19 on social interaction, lower perceived social support, more severe academic/occupational interference due to COVID-19, lower COVID-19-specified support, and poorer self-reported physical health were significantly associated with sleep disturbance. Less handwashing, lower perceived social support, lower COVID-19-specified support, poorer self-reported physical health, and younger age were significantly associated with suicidal thoughts.

### 4.1. Worry about COVID-19

The present study found that a high level of worry about COVID-19 was significantly associated with sleep disturbance. Previous study indicated that degree of worry was significantly associated with psychological distress measured by General Health Questionnaire-28 during 2009 A/H1N1 influenza [18]. Another study regarding 2015 Middle East Respiratory Syndrome (MERS) reported that individuals with very high stress in daily life had higher levels of worry than those who reported having little stress [19]. People with significant psychological distress may suffer from sleep disturbance. In addition, it had been reported that misinformation about COVID-19 has been proliferating on social media [20,21]. Owing to the significant association between worry about COVID-19 and sleep disturbance, whether misinformation on social media may deepen people’s worry about COVID-19 warrants further investigation.

### 4.2. Changes in Social Interaction and Perceived Support during the COVID-19 Pandemic

Social distancing is one of the main protective behaviors against contracting COVID-19 [22]. Moreover, previous studies have found that the Severe Acute Respiratory Syndrome (SARS) outbreak changed social interaction because of the fear of contracting the disease among the public [23] and due to healthcare workers being stigmatized [24]. The present study found a significant association between changes in social interaction due to COVID-19 and sleep disturbance, but not significant for suicidal thoughts. Instead, the present study found that lower perceived social support and specific support against COVID-19 were both independent predictors for sleep disturbance and suicidal thoughts. Previous research has found that perceived support from a family member, friends, and medical staff were associated with mental health during the SARS pandemic [25], and also that insufficient social support was a risk factor for depression, anxiety, and sleep problems among healthcare workers in the COVID-19 pandemic [26]. The results of this and previous studies demonstrate the importance of sufficient social support and specific support against infective respiratory diseases during a pandemic. Social distancing may hamper physical, social interaction; however, social support can be offered by telecommunication. Governments should provide support for those who are socially isolated before the pandemic and for those who are quarantined due to infection during the pandemic to help prevent mental health problems.

### 4.3. Interference with Academic/Occupational Performance and Lifestyles Due to COVID-19

To prevent the spread of COVID-19, authorities all over the world announced stay-at-home and school-closure orders, which interfere with people’s academic/occupational performance. The present study found that the higher the academic/occupational interference by COVID-19 the higher the predicted sleep disturbance. Although academic and occupational activities may be restored as the risk of COVID-19 is mitigated, it takes time and consistent policies to restore economic prosperity. The unemployment rate increased because of the enormous impact of COVID-19 on the global economy. Research has found that the unemployment rate increased in parallel with the prevalence of depressive and anxiety disorders [27]. Academic and occupational interference by the COVID-19 pandemic and the negative effect this has on health should be monitored not only during but also after the pandemic.

The daily lives of the public have changed due to the adoption of protective behaviors against contracting COVID-19 and searching for information on COVID-19. The present study found that less handwashing was significantly associated with suicidal thoughts in the multiple logistic regression. Handwashing is the most recommended behavior by the WHO to protect individuals from contracting COVID-19 [22]. However, people who had suicidal thoughts might not have the motivation to adopt the experts’ recommendation of hand washing. This association may be influenced by level of depression. Previous study revealed the association between depression prevalence rates and poor health habits [28]. In addition, people with mental illnesses have a greater vulnerability to serious complications from COVID-19 due to cognitive impairment, little awareness of risk, and diminished efforts regarding personal protection [29]. Early intervention for mentally ill individuals during the COVID-19 pandemic is crucial for preventing a possible loophole for infection control.

### 4.4. Self-Reported Physical Health in the COVID-19 Pandemic

The present study found that poorer self-reported physical health was significantly associated with sleep disturbance and suicidal thoughts. Several possible etiologies may account for this association. First, poorer physical health, such as chronic diseases or impaired physical function, may directly increase psychological distress [30]. Second, people might stop seeing the doctors because they were worried about contracting COVID-19 and the treatment for physical problems might be delayed. Both anticipatory anxiety about and actual occurrence of exacerbated illnesses may result in sleep disturbance and suicidal thoughts. Third, poorer physical health may result in reduced physical activities, which was reported to be associated with higher psychological distress [31]. Fourth, sleep disturbance may worsen physical health [32]. Moreover, people with suicidal thoughts were likely to have depressive disorders, which were significantly associated with physical health problems [33]. Although the cross-sectional study design limited the ability to determine the causal relationship between self-reported physical health, sleep disturbance, and suicidal thoughts, health professionals should encourage the public to continuously enhance physical health in addition to detecting the symptoms of COVID-19 infection.

### 4.5. Demographic Factors Associated with Sleep Disturbance and Suicidal Thoughts

Younger age was an independent predictor for suicidal thoughts as determined by multivariate logistic regression. It was also potentially associated with sleep disturbance. A recent study in China also demonstrated that younger subjects were at high risk of mental illnesses, such as general anxiety disorder during the COVID-19 pandemic [6]. Moreover, the present study found that non-healthcare workers and transgender people were potentially more likely to have suicidal thoughts than healthcare workers and females, respectively, as determined by univariate logistic regression. However, these associations became insignificant in the multiple logistic regression. The study in China found that healthcare workers were at high risk for poor sleep quality during the COVID-19 pandemic [6], whereas the present study did not find more severe sleep disturbance in healthcare workers compared with non-healthcare workers. Given that the COVID-19 epidemic in China was much more severe than in Taiwan [4], we suggest that varying healthcare burdens across the COVID-19 pandemic may account for the discrepancy between the results of the present and other studies. Transgender people were reported to have higher rates of depression, suicidal thoughts, and other mental illnesses compared with other genders [34]. The psychological impact of infectious disease pandemics on gender minorities should not be neglected.

### 4.6. Limitations

The present study has several limitations. First, although recruiting participants through the internet is a promising research method for targeting the general public, possible selection bias exists for those who are not netizens. Hence, a paper-and-pencil test and advertisements posted on public area may recruit participants who rarely surf the internet. Second, the cross-sectional design of this study during acute pandemic stage limited causal inference and the presence of sufficient information at follow-up. A prospective follow up study may further explore the causality between variables and change of measurements with time, especially during plateau or remission stage. Third, this study was conducted during the period of COVID-19 mitigation but not during period when COVID-19 first emerged in Taiwan. Therefore, the initial impact of COVID-19 cannot be identified. Finally, sleep pattern and suicidal thoughts were only measured by a single question, which may limit the implication of the current study. Formal questionnaires such as the Pittsburgh Sleep Quality Index may provide comprehensive assessment for sleep problems.

## 5. Conclusions

The present study identified several COVID-19-related predictors for sleep disturbance and suicidal thoughts among people during the COVID-19 pandemic. This massively undesirable effect on mental health deserves more attention and support from authorities during the COVID-19 pandemic. For instance, a lower support system was associated with higher risk of sleep disturbance and suicidal thoughts, indicating the importance of timely support during pandemics. In addition, the significant association between less handwashing and suicidal thoughts demonstrated the possibility that individuals with suicidal thoughts may have poor coping strategies of infection control. Hence, intervention to those with mental health problems during the COVID-19 outbreak is necessary to enhance their coping with the threats of infection, which may decrease the risk of poor infection control. The study provides information that could aid timely intervention in the mental health of the public. Moreover, sleep disturbance and suicidal thoughts may occur not only during the COVID-19 pandemic but also during the restoration period. Further investigation is needed to fill the limitations of the current study. The prospective study with follow up at different stages of the pandemic can help us better understand the changes of association between mental health and multi-dimensional factors. Moreover, studies with multiple recruiting sources and detailed questionnaires of mental health problems can further extend the application of the findings of the current study.

## Figures and Tables

**Table 1 ijerph-17-04479-t001:** Associations of demographic factors with sleep disturbance and suicidal thoughts examined by univariate logistic regression (*n* = 1970).

			Sleep Disturbances				Suicidal Thoughts			
	Mean	SD	B	cOR	95% CI	*p*	B	cOR	95% CI	*p*
Age (years)	37.81	11.00	−0.01	0.99	0.979–0.995	**0.002**	−0.07	0.93	0.91–0.94	**<0.001**
	*n*	%	B	OR	95.0% of CI	*p*	B	OR	95% CI	*p*
Gender										
Female	1305	66.2	-	-	-	-	-	-	-	-
Male	650	33	−0.11	0.90	0.74–1.09	0.270	−0.06	0.94	0.69–1.28	0.697
Transgender	15	0.8	0.14	1.15	0.41–3.25	0.793	1.71	5.55	1.95–15.82	**0.001**
Education level										
High school or below	218	11.1	-	-	-	-	-	-	-	-
College or above	1752	88.9	0.08	1.08	0.81–1.43	0.601	0.03	1.03	0.65–1.62	0.915
Healthcare workers										
No	1324	67.2	-	-	-	-	-	-	-	-
Yes	646	32.8	−0.07	0.93	0.77–1.13	0.476	−0.88	0.41	0.29–0.60	**<0.001**

CI = confidence interval; cOR = crude odds ratio; SD = standard deviation; bold = *p* < 0.05.

**Table 2 ijerph-17-04479-t002:** COVID-19 related factors associated with sleep disturbance and suicidal thoughts examined by univariate logistic regression (*n* = 1970).

			Sleep Disturbances				Suicidal Thoughts			
	**Mean**	**SD**	**B**	**cOR**	**95% CI**	***p***	**B**	**cOR**	**95% CI**	***p***
Worry about COVID-19	19.66	4.78	0.07	1.07	1.05–1.09	**<0.001**	0.03	1.03	1.00–1.07	**0.032**
Change in social interaction due to COVID-19	1.35	1.42	0.13	1.14	1.07–1.22	**<0.001**	0.02	1.02	0.92–1.12	0.729
Academic/occupational interference by COVID-19	1.67	1.18	0.18	1.20	1.11–1.29	**<0.001**	0.11	1.12	0.99–1.26	0.072
Perceived social support	8.59	2.01	−0.14	0.87	0.83−0.91	**<0.001**	−0.32	0.73	0.68–0.79	**<0.001**
Specific support against COVID-19	8.80	1.73	−0.17	0.85	0.80–0.89	**<0.001**	−0.27	0.76	0.71–0.82	**<0.001**
Self-reported physical health	4.15	1.59	−0.27	0.76	0.72–0.81	**<0.001**	−0.33	0.72	0.65–0.79	**<0.001**
	***n***	**%**	**B**	**OR**	**95.0% CI**	***p***	**B**	**OR**	**95% CI**	***p***
Change in lifestyle due to COVID-19										
Avoiding crowded places										
No	111	5.6	-	-	-	-	-	-	-	-
Yes	1859	94.4	0.15	1.16	0.79–1.71	0.441	−0.43	0.65	0.38–1.11	0.648
Indoor ventilation										
No	232	11.8	-	-	-	-	-	-	-	-
Yes	1738	88.2	−0.25	0.78	0.59–1.03	0.077	−0.05	0.95	0.61–1.47	0.816
Household disinfection										
No	657	33.4	-	-	-	-	-	-	-	-
Yes	1313	66.6	0.21	1.23	1.02–1.49	**0.030**	−0.28	0.75	0.56–1.01	0.059
Handwashing										
No	164	8.3	-	-	-	-	-	-	-	-
Yes	1806	91.7	0.31	1.36	0.99–1.87	0.061	−0.74	0.48	0.31–0.72	**0.001**
Wearing a mask										
No	213	10.8	-	-	-	-	-	-	-	-
Yes	1757	89.2	0.23	1.26	0.95–1.67	0.114	−0.35	0.71	0.47–1.07	0.099
Acquiring knowledge of COVID-19										
No	467	23.7	-	-	-	-	-	-	-	-
Yes	1503	76.3	0.18	1.16	0.97–1.47	0.098	−0.08	0.92	0.66–1.29	0.639
Intentionally miss clinic reservation										
No	1642	83.4	-	-	-	-	-	-	-	-
Yes	328	16.6	0.08	1.08	0.85–1.37	0.541	−0.21	0.81	0.54–1.21	0.302

CI = confidence interval; cOR = crude odds ratio; SD = standard deviation; COVID-19 = Coronavirus disease 2019; bold = *p* < 0.05.

**Table 3 ijerph-17-04479-t003:** Distribution of sleep disturbance and suicidal thoughts (*n* = 1970).

Dependent Variables	Sleep Disturbance		Suicidal Thoughts	
Distribution	*n*	%	*n*	%
Never	871	44.2	1758	89.2
Mild	774	39.3	129	6.5
Moderate	232	11.8	57	2.9
Severe	79	4.0	17	0.9
Extremely severe	14	0.7	9	0.5

**Table 4 ijerph-17-04479-t004:** Predictors of sleep disturbance and suicidal thoughts examined using forward stepwise logistical regression.

***Predictors of Sleep Disturbance***	**B**	**aOR**	**95% CI**	***p***
Worry about COVID-19	0.04	1.04	1.02–1.06	**0.001**
Academic/occupational interference by COVID-19	0.12	1.12	1.04–1.22	**0.005**
Perceived social support	−0.10	0.91	0.87–0.96	**<0.001**
Specific support against COVID-19	−0.09	0.92	0.86–0.97	**0.003**
Self-reported physical health	−0.22	0.80	0.76–0.86	**<0.001**
Change in social interaction due to COVID-19	0.07	1.07	1.00–1.15	**0.047**
***Predictors of suicidal thoughts***	**B**	**OR**	**95% CI**	***p***
Age	−0.07	0.93	0.92–0.95	**<0.001**
Handwashing	−0.74	0.48	0.30–0.76	**0.002**
Perceived social support	−0.24	0.79	0.73–0.85	**<0.001**
Specific support against COVID-19	−0.21	0.81	0.75–0.88	**<0.001**
Self-reported physical health	−0.21	0.81	0.73–0.90	**<0.001**

CI = confidence interval; aOR = adjusted odds ratio; SD = standard deviation; COVID-19 = Coronavirus disease 2019; bold = *p* < 0.05.

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
