# Peer review of "COVID-19-Related Factors Associated with Sleep Disturbance and Suicidal Thoughts among the Taiwanese Public: A Facebook Survey"

_ijerph, 2020, doi:10.3390/ijerph17124479_

Round 1

Reviewer 1 Report

The manuscript is well-written and well-organized. Authors did a great job. I recommend manuscript for publication however I suggest the following revisions to be addressed before I endorse manuscript for publication. 

Line 48: Please make sure you also add the year after March as well.

Line 56: what do you mean that studies are “insufficient”? In terms of results, outcomes? Please specify. To the best of my knowledge thousands of papers have been published in regards to COVID so you may need to cite some of those studies to justify your argument or at least to publish some major organization’s argument about this issue as well. Or if it is your personal opinion also state it in the paper.

Line 65: you do not need a comma after 18.2%

Line 81: What do you mean by “COVID-19-related factors associated with…” ? I think this sentence is vague unless you would like to name those particular factors you are referring to. Otherwise I would delete this sentence, maintain following sentence “We are particularly interested in….” and I would add the sleep disturbance and suicidal thoughts to that sentence….that way your study aims are delivered clearly.

Lines 107-113: So you only used one item to measure sleep disturbance and suicidal thoughts. My opinion is that those issues are quite complex and one item is not going to clarify different angles of it. So, I suggest that you identity that as one of your limitations at the end of your manuscript.

Please expand in your conclusion section and add more information about clinical, policy, future research implications based on your study findings.

Author Response

As attached file

Reviewer 2 Report

I feel the study is relevant and well conducted. Also it has been presented in a very clear way. I was surprised by the possible link between

 Less handwashing; lower perceived social support;  lower COVID-19-specified support; poorer self-reported physical health; and younger age were significantly associated with suicidal thoughts. 

You say

However, people who had suicidal thoughts might not have the 278 motivation to adopt the experts’ recommendation of hand washing. 

Are there other studies which show a link between suicidal thoughts and people who do not look after their physical health.

Author Response

as attached file

Reviewer 3 Report

Language improvements

In general : Check entire for incorrect use of semicolon instead of comma, and inappropriate use of commas/ Write as much as possible in third person singular, and avoid first person plural (we)

P1/L 24 - remove 'the'
p1/L26 - replace 'from' with 'through'
P1/26 - replace 'this (a survey cannot recruit participants by itself'
P1/L28 remove , before along.
P1/L29- remove 'their'.
P1/L32 - Replace 'a more severe' for 'increased'
P1/L37. Multiple COVID-19-related factors were associated with sleep disturbance and suicidal thoughts in the COVID-19 pandemic
This sentence should appear earlier in the manuscript

P1/L38. Timely intervention and sufficient support are important for the public during the COVID-19 pandemic.
This is not a conclusion of the results. Ommit.

1.2
In general: The entire section is badly structured. It jumps from COVID-19 to Sars, than back to COVID-19 again, than mentions it is import to study factors, than to SARS and Covid again. Please restructure adequately.

P2/L51. What are knock-on effects?
P2/L52. Please specify the heavy burden on society.  suggest adding more information on COVID-19 in the context of Taiwan.
Who many live in people? When did it emerge? How did it develop? What was the reaction of people? How many people where infected? How many died? What were the main characteristcis of the deceased (Age, condition)? How did healthcare institons handle the situation? Where there enough intense care facilities. What measures were taken? Did the population accept these measures?
etc. etc!

P2/L56- Although there a a limited number of studies on the effects of COVID, there are some paper. Please report findings from these studies (Tip: search Researchgate and Academia.edu for recent studies)

1.3 I suggest opening this section with the line that indicates why more research is important. (Lines 67 -70)

2.2 Questionnaires (plural)
In general: Reporting is confusing and inconsistent. Most sections mention the exact questions, others the themes. Sometimes Cronback Alpha is mentioned, sometimes not. I suggest a consistent way of reporting. The themes of the questions could be described, along with the Lickert scale range and Cronbach Alpha. The exact questions can be added tot the paper as an Appendix/ Supplemental material. All numbers below 10 should be written out in word form.

2.2.2 Demographic variables
Why is this mentioned secondly? Should be 2.2.1

2.2.4
With reference to a previous study - What does this mean?

3.1 'A total of 1,099 (55.8%) participants reported sleep 186 disturbance, and 212 (10.8%) reported having suicidal thoughts in the past week'.This is an outcome, not a descriptive Reporting is not logical.I suggest:
3.1 Descriptive analysis
3.2 Sleep disturbance and suicidal ideation

4
In general: this section is badly structured, confusing to read. Please start with a description with the outline of the discussion. After that start with summarizing the findings of the primary outcome: Sleep disturbance and suicidal ideation.  Then describe main findings of the secondary outcomes: predictors

Findings of other studies should be described more detailed. Just mentioning that 'research has also found' is not good enough
First report own findings, then compare to other studies, not reversed!
'
This massively undesirable effect on mental health deserves more attention and support from authorities during the COVID-19 pandemic/ .
Why? I suggest moving this sentence to conclusion, it is inappopriate here

4.6 Limitations
This section is too short. What are the possible ramifications of the limititation (refer to other studies). What can be done to counterpart the limitations.

What are practical implications of this study?
What are suggestions for future research?

Author Response

as attached file

Round 2

Reviewer 3 Report

Dear authors 

I would like to thank you for the quick revision, which I found solid.

The findings of the studies are very interesting. I do not know what is more alarming, the fact that such a high percentage of the population reports sleep disturbances and suicidal thoughts (56% and 11%), or  that in Taiwan which has a population of nearly 24 million residents, only 7 people have died from corona.

I wish you luck on further research.

Best regards
dr. Tom Hendriks